# 2D Numerical Simulation of Auxetic Metamaterials Based on Force and Deformation Consistency

**DOI:** 10.3390/ma15134490

**Published:** 2022-06-25

**Authors:** Antonina Roth, Georg Ganzenmüller, Florian Gutmann, Puneeth Jakkula, François Hild, Aron Pfaff, Kaiyang Yin, Chris Eberl, Stefan Hiermaier

**Affiliations:** 1INATECH, Albert-Ludwigs-Universität Freiburg, Emmy-Noether-Straße 2, 79110 Freiburg, Germany; georg.ganzenmueller@inatech.uni-freiburg.de (G.G.); florian.gutmann@emi.fraunhofer.de (F.G.); puneeth.jakkula@inatech.uni-freiburg.de (P.J.); stefan.hiermaier@emi.fraunhofer.de (S.H.); 2Fraunhofer Institute for High-Speed Dynamics, Ernst-Mach-Institut, EMI, Ernst-Zermelo-Straße 4, 79104 Freiburg, Germany; aron.pfaff@emi.fraunhofer.de; 3University Paris-Saclay, CentraleSupélec, ENS Paris-Saclay, CNRS, LMPS-Laboratoire de Mécanique Par-is-Saclay, 4 Av. des Sciences, 91190 Gif-sur-Yvette, France; francois.hild@ens-paris-saclay.fr; 4IMTEK, Albert-Ludwigs-Universität Freiburg, Georges-Köhler-Allee 078, 79110 Freiburg, Germany; kaiyang.yin@imtek.uni-freiburg.de (K.Y.); chris.eberl@iwm.fraunhofer.de (C.E.); 5Fraunhofer Institute for Mechanics of Materials IWM, Wöhlerstrasse 11, 79108 Freiburg, Germany

**Keywords:** additive manufacturing, metals, lattice structure, auxetic, simulation model, DIC

## Abstract

This work showcases a novel phenomenological method to create predictive simulations of metallic lattice structures. The samples were manufactured via laser powder bed fusion (LPBF). Simulating LPBF-manufactured metamaterials accurately presents a challenge. The printed geometry is different from the CAD geometry the lattice is based on. The reasons are intrinsic limitations of the printing process, which cause defects such as pores or rough surfaces. These differences result in material behavior that depends on the surface/volume ratio. To create predictive simulations, this work introduces an approach to setup a calibrated simulation based on a combination of experimental force data and local displacements obtained via global Digital Image Correlation (DIC). The displacement fields are measured via Finite Element based DIC and yield the true local deformation of the structure. By exploiting symmetries of the geometry, a simplified parametrized simulation model is created. The simulation is calibrated via Response Surface Methodology based on nodal displacements from FE-DIC combined with the experimental force/displacement data. This method is used to create a simulation of an anti-tetrachiral, auxetic structure. The transferability and accuracy are discussed, as well as the possible extension into 3D space.

## 1. Introduction

This work showcases a novel phenomenological method to create predictive simulations of auxetic metamaterials made from metal using laser powder bed fusion (LPBF). In the context of this work, the term metamaterial is used to describe a structured material whose properties are very different from the intrinsic properties of the constituting base material used for manufacturing. The mechanical properties of the metamaterial are governed by its geometrical structure on a length scale smaller than the overall dimensions of the overall manufactured object, i.e., its external dimensions, but larger than the microscopic length scales, i.e., the atomistic structure. As this work considers a lattice structure, this new length scale is the size of the lattice unit cell.

Auxetic structures exhibit a well-defined negative Poisson’s ratio [1]. The auxetic effect is a structural response facilitated by the geometry of the unit cell. The base for the auxetic effect is local rotations of structural elements. Compared to conventional lattice structures, metallic auxetic metamaterials exhibit increased yield strength, energy absorption [2], and energy dissipation [3], especially at high rates of strain. The extensive elastoplastic deformation of the internal structure yields a characteristic stress plateau at increasing strains and, therefore, enhanced energy absorption capabilities [4,5,6]. Predominant bending deformation, as seen in chiral structures specifically, provides an even higher energy absorption potential [7,8]. This feature makes auxetic materials interesting for applications in, e.g., the automotive industry, specifically in the crash sector.

Simulations of LPBF-manufactured metamaterials based on nominal, i.e., construction geometry from the design stage, present a challenge. The reason is the difference between the printed geometry and its nominal CAD configuration (Figure 1) [9]. In addition, the printing process induces defects such as porosity, and the behavior of the base material is different in bulk compared to a slender structure due to surface effects [10,11,12,13]. Therefore, simple models do not yield realistic results, especially when non-linear behavior is observed, e.g., plasticity [14,15]. Furthermore, simulation predictions tend to focus mostly on the strength and energy absorption of the structure while not capturing the deformation over time in detail [16,17,18]. This limitation leads to simulations that are close to their experimental equivalent but do not necessarily replicate the overall structural behavior or the detailed kinematics of individual parts. 

Metamaterial simulations often employ a discretization using 3D elements. The discretization is either based directly on the structure obtained via CT imaging [19,20,21], resulting in a model that captures all defects, or the CAD model [18,22], resulting in an ideal model that neglects the effect of manufacturing imperfections. While both describe the complex geometry in certain detail, the resulting simulations are computationally expensive due to high mesh density. Simulations in applications that require extensive numerical studies, such as machine learning, result in long calculation times and call for extensive resources. 

In order to circumvent this issue, beam elements are used to discretize periodic lattice structures [23,24]. Beam models require fewer resources in comparison to full 3D models while having the same potential to predict material properties [25,26]. Potential irregularities due to the manufacturing process are either omitted, implemented by varying the cross-section of the beam [24,27], or accounted for by assigning different mechanical properties to each strut [28]. Therefore, extensive experimental characterization is necessary in order to represent manufacturing defects using beam elements.

Thus, a new approach to solve this issue is needed. The goal here is to design a simulation that is not only predictive but also numerically efficient. The idea is to create a tool for engineers and scientists to setup a reliable simulation with little experimental characterization in order to be able to numerically analyze material kinematics and realistically predict structural behavior in different contexts.

A new approach is introduced to setup metamaterial simulations based on a combination of force and deformation consistency. The method is demonstrated using an anti-tetrachiral auxetic structure. This structure is additively manufactured and experimentally characterized using a compressive load. The experiment was analyzed via global Digital Image Correlation (DIC). By exploiting symmetries of the geometry, a simplified parametrized simulation model is created. The simulation model was calibrated based on the data acquired in the experiment, thereby resulting in an efficient simulation. The prediction capabilities of the simulation framework are discussed by investigating the simulation under varying loading and geometrical conditions. Furthermore, the work discusses the applicability to other geometries and the extension in the third dimension.

## 2. Materials and Methods

This section introduces the structure analyzed in this work, including the manufacturing process, the base material, and the experimental setup. Furthermore, the section establishes the approach used to analyze the experiments and setup a simulation based on the corresponding data.

### 2.1. Manufacturing and Experimental Setup

The chosen metamaterial is an anti-tetrachiral structure. The geometry is based on the structures shown in Ref. [29]. This structure was selected due to its auxeticity and low shear modulus, making it a scientifically interesting metamaterial to analyze. The two-dimensional structure is extruded into the third direction in space, creating a three-dimensional CAD model suitable for printing. 

Specimens based on the dimensions depicted in Figure 2 were printed using laser powder bed fusion (LPBF). LPBF builds the structure layerwise from fine metal powder. Each layer of powder is distributed onto a substrate plate and selectively molten with a laser, creating one layer of the structure. The laser is powerful enough to fully melt the powder and create a solid material. The process is repeated until the structure is complete [30]. 

The manufacturing was executed by Fraunhofer Ernst-Mach-Institute (Fraunhofer EMI) using an EOS M 400 3D printer. Scalmalloy^®^ was the material of choice, which is a Scandium modified Aluminium-Magnesium alloy. Scalmalloy^®^ stands out due to its increased tensile strength and failure strain. Both properties are similar to the characteristics of some titanium alloys while featuring a lower mass density. Scalmalloy^®^ was created as a base for lightweight crash-absorbing structures, specifically for the automotive sector and aerospace engineering [31]. Due to its ductility, it can sustain extensive plastic deformation before failure. This material property allows one to take advantage of the full structural deformation potential before failure. The material was not heat-treated after the printing process. 

The structure was positioned parallel to the base plate of the printer, with each additional layer extruding the geometry into the third direction in space. This choice ensures identical material properties in all struts. The relative mass density of the 3D-printed metamaterial with respect to the mass density of the base material, Scalmalloy^®^, is 29.5%, considering the enclosing cube of 27 × 27 × 27 mm^3^ as the reference volume.

In order to setup the simulation, experimental data are needed. For mechanical testing, the structure is compressed at a strain rate of 10^−3^/s using a Zwick Z100 testing machine in displacement control mode. The force was measured using a 100 kN load cell with an accuracy better than 0.1% for forces greater than 200 N. The structural deformation is recorded as monochrome images using a Basler ace 2 camera. Approximately 250 images with a pixel resolution of 11.3 μm are acquired per experiment. Images are then post-processed via Digital Image Correlation to yield a quantitative full-field displacement measurement.

### 2.2. Digital Image Correlation

A quantitative analysis of the structural response creates a base for in-depth understanding of the structural behavior and provides data to setup a simulation of the metamaterial. An optical method to conduct such an analysis is Digital Image Correlation (DIC). DIC computes a displacement field by registering two or more images.

Since the images are divided into sub-elements in the process, and the goal is to extract kinematic details, DIC is combined with a finite element mesh based on the nominal CAD geometry. The result is called Finite Element based DIC. The method used in this work was shown in Ref. [32], applied to pantographic structures. A mesoscale mesh is constructed, replicating the nominal CAD geometry. Based on the mesh and the position of the structure within the image, a mask is created. The gray level of the mask is equal to the average gray level of the structure. Due to the uniform background, the mask is backtracked to the initial image, adjusting the position and orientation of the mesh relative to the structure [33]. This yields the reference configuration. A direct calculation is followed as the registration route, where every nth picture is registered with the initial image. The result is a displacement field, which allows kinematic details ro be revealed with the deformed structure and therefore allows its mesoscale behavior to be analyzed, as shown in Figure 3.

In this work, the displacement field was obtained using Correli 3.0 [34]. The calculation utilizes regularized FE-based global DIC [35,36]. All hardware and software parameters are summarized in Table 1. The usage of global DIC is critical in the calibration process. In comparison to local DIC, global DIC can utilize regularization, which yields a kinematically correct displacement field.

### 2.3. Finite Element Simulation

Compared to other model calibration methods, the goal of this work is not the calibration of a material model. The bulk material behavior is assumed. The material model is calculated from a separate set of quasi-static tensile tests. The specimens were manufactured using Scalmalloy^®^. The corresponding data are provided in Ref. [31]. In this work, a power-law elastoplastic model was employed. 

Since the structure is a 2D geometry extruded in the third direction in space, it suggests considering a quasi-2D discretization. This remark raises the question of whether a plane strain or plane stress simulation would be appropriate. It was established, both experimentally and numerically, that a 2D plane strain simulation approach is sufficient. A corresponding simulation model is set up by analyzing the cross-section of a single LPBF printed beam and exploiting symmetries of the geometry. The finished model was calibrated using Response Surface Methodology.

#### 2.3.1. Model Setup

The focus of this work is the influence of geometrical variation on structural deformation and strength. As depicted in Figure 4a, the cross-section of an LPBF printed metallic structural component typically consists of a core of solid material, which is encased by an additional uneven surface layer. Further analysis reveals (see Figure 4b) that a big part of this outer layer consists of metal particles that were not completely molten as the rest of the solid metal. Furthermore, an analysis of the grain orientation of a round specimen shows a clear distinction between a mostly homogeneous core and a heterogeneous, rough outer layer.

The rough outer layer does not contribute significantly to the mechanical stiffness or strength of the strut because it does not represent a continuous cross-section that may carry the mechanical load. Thus, the stiffness and strength are mostly a function of the continuous core cross-section only. However, the rough surface contributes to the force response at high compression ratios–within the densification regime. Here, struts come into contact with each other, so the outer surface geometry becomes important. This observation allows the strut to be divided into two components, which have to be numerically defined. Beam elements offer the simplest approach to replicating the composition of the strut. They are a numerically efficient alternative to 3D elements and are used for a variety of structures, especially if the structure is periodic. The stiffness and strength of the solid core are defined by assigning a beam thickness to each node. The values between two neighboring nodes are interpolated to ensure continuity. Each beam thickness is geometrically smaller than the total size of the corresponding printed strut. The lattice structure is subjected to large global deformation. In order to simulate the subsequent onset of densification, the simulation model requires an equivalent to the surface layer. As the influence of the surface layer is only significant in the case of self-contact, it is represented by a contact offset. Besides defining the onset of densification, contact offset does not impact other physical properties of the beam.

The complexity of the model is reduced by exploiting the symmetries of the structure. The C_4_-rotational symmetry of the unit cell combined with the reflective symmetry reduces the parameters needed to only two beam thicknesses and one contact offset (see Figure 5).

In order to directly compare the deformation of the beam model with the DIC results, the DIC mesh is overlaid with the beam mesh. Using neighboring functions, each node of the beam mesh gets assigned its respective equivalents within the DIC mesh. Depending on the element size of the DIC mesh, this can be either one or multiple nodes. This relation allows the target displacements to be calculated for each node within the beam model, forming a target database of nodal displacements to be used in the following calibration. Furthermore, the beam model is aligned with the position of the structure within the image.

Several experiments and simulations of different structures have shown the force responses of the structures to be significantly dependent on their boundary conditions. In order to replicate the experimental results as close as possible, displacements obtained from DIC are utilized. Nodes on the top and bottom of the lattice are defined as boundary nodes. Two prescribed motions are assigned to each boundary node in the x and y directions. The rest of the nodes can move freely, except for the constraint that they cannot penetrate into the compression platens of the testing machine.

The setup is simulated in LS DYNA using an implicit solver. The simulation consists of 288 beam elements with a 2D plane strain element formulation and an average element size of 1.5 mm. In plane strain, strains in the third direction in space are assumed to be negligible in comparison to those in the other two directions [37]. It was checked experimentally and numerically that the strain in the direction of extrusion is 100 times less than that in the corresponding loading direction, only increasing beyond the onset of densification.

A parametric study is conducted in order to analyze the influence of each parameter on the simulation. The first beam thickness (beam thickness 1), also referred to as the outer beam thickness, is the primary parameter that defines the stiffness and strength of the structure. Values lower than 0.2 mm result in buckling effects, while higher values do not significantly influence the deformation field. 

The second beam thickness, also called inner beam thickness, defines the shear stiffness of the structure. The influence of this parameter is partly dependent on the outer beam thickness. If the value of the inner beam thickness becomes lower than 40% of the outer beam thickness, the structure tends to bend laterally instead of showing an auxetic behavior. For values higher than 40%, no effect of the second beam thickness is observed.

The third parameter, the contact offset, only influences the behavior of the structure when it experiences self-contact, i.e., in the densification regime at compressive strains beyond 20%. The value is defined as a one-sided offset starting at the center of the beam. In order to obtain the total size of the beam, the value is doubled. Thus, twice the contact offset should always be greater than either outer or inner beam thicknesses (see Figure 4a). The contact offset defines the onset of densification and therefore influences both the force progression and the deformation at high values of compressive strain.

#### 2.3.2. Model Calibration

To obtain the model parameters, an optimization approach termed Response Surface Methodology (RSM) [38] is used in combination with sequential domain reduction. The method was originally developed for optimal sampling of physical experiments but has been adapted quickly to numerical applications due to its robustness. Here, it is used to calculate the optimum geometrical model parameters. The optimum is defined as those parameters that minimize the deviation between experimental measurements and simulation results. The metric employed to quantify this deviation is the Normalized Root Mean Squared Error (NRMSE) [39,40]. An in-house code in MATLAB was employed to carry out the task.

The script utilizes a multivariable and constrained optimization approach. After defining the parameters and their respective domains, the parameter space is sampled to create an initial guess. A refinement towards the optimum is iteratively obtained by changing the parameters according to the methods of Central Composite design CCD [41] and D-optimal design [42] so as to minimize the total NRMSE.

This error has two contributions: the force and displacement error. To sum up these two, they are first rendered dimensionless by normalizing with respect to the range of forces encountered in the experiment. The Mean Square Error ${\mathrm{MSE}}_{F}$ is divided by the difference between the maximum target value $T_{F}{}_{\max}$ and minimum target value $T_{F}{}_{\min}$, obtaining the normalized force error NRMSE_F_:(1)$${\mathrm{NRMSE}}_{F}=\sqrt{{\mathrm{MSE}}_{F}}/(T_{F}{}_{\max}-T_{F}{}_{\min})$$

In the case of the normalized nodal displacement error, NRMSE_D_, a scalar measure is needed from the displacement field difference between experiment and simulation. To this end, dimensionless root-mean-square averages are calculated along the Cartesian directions, x and y, separately. They are then normalized by the number of nodes m, where the displacement is evaluated. Thus the average error over all nodes is obtained: (2)$${\mathrm{NRMSE}}_{D}=\frac{1}{2m}\overset{m}{\underset{j=1}{\sum }}{\left[\frac{\sqrt{{\mathrm{MSE}}_{\mathrm{XD}}}}{{(T}_{\mathrm{XD}}{}_{\max}-T_{\mathrm{XD}}{}_{\min})}+\frac{\sqrt{{\mathrm{MSE}}_{\mathrm{YD}}}}{{(T}_{\mathrm{YD}}{}_{\max}-T_{\mathrm{YD}}{}_{\min})}\right]}_{j}$$

Force and displacement errors are combined using a weighting factor, a, resulting in the total error of the system. In this work, a = 0.5 is selected:NRMSE = a · NRMSE_F_ + (1 − a) · NRMSE_D_(3)

The NRMSEs for a particular set of parameter values are utilized to build the response surface, which is approximated using a quadratic polynomial. By locating the minimum of the response surface, a subdomain is created [43]. This subdomain is used to calculate new sets of parameters and run new simulations, thus iteratively converging to an optimum. In the present implementation, the optimization is deemed converged after either the domain changes of all parameters or the change of the NRMSE reach 10^−3^ or less.

## 3. Results and Discussion

This section visualizes and discusses calibration results both qualitatively and quantitatively. Furthermore, the calibrated parameter set is applied to similar simulations with different loading situations in order to analyze its transferability.

### 3.1. Calibration Results

As depicted in Figure 6, the simulation managed to reproduce the deformation field well. Note that the quantity “compressive strain” refers to the change in sample height normalized by the initial sample height. The maximum displacement error of any single node was less than 5%. Additionally, the simulated force response agreed well with the experimental data, including beyond the point of densification. Up to a compressive strain of 24%, the force error was calculated as 3.4%. Furthermore, the simulation required only a fraction of simulation time and computational resources than more advanced models. Compared to a model discretized using pentahedra elements with a size of 0.27 mm, the calibrated simulation required a factor of 10^5^ less time to run, terminating after 9 s, rather than 10 days using one CPU core.

Comparing all resulting parameter values (beam thickness 1 and 2 and contact offset) to each other, the total contact offset was larger than both beam thicknesses. This result confirms the observation in Section 2.3.1 as the total size of the strut always has to be larger than its solid core. Furthermore, the average strut thickness was measured as 0.64 mm. The total contact offset was only 1.9% smaller than the measured value, excluding arbitrariness and confirming the physical significance of the calculated parameters. Detailed calibration results are reported in Table 2.

### 3.2. Validation

This section analyses whether the calibrated parameters obtained for one unit cell geometry and a specific loading situation can be transferred to similar but different situations. In the first investigated case, the unit cell geometry was kept the same, but the outer dimensions of the sample and the boundary conditions were changed. Following this, a case where the unit cell was similar in many dimensions but had different symmetry elements was investigated. 

#### 3.2.1. Transferability to Different Loading Conditions

This section analyses if the calibrated simulation parameters can be transferred to different loading conditions and different macroscopic sample geometries. The unit cell itself is kept the same, but the outer dimensions of the simulation sample are changed.

Two different samples were manufactured, as shown in Figure 7. The geometry of the unit cell was not changed and is shown in Figure 2. Both experiments were analyzed using FE-based DIC in order to obtain the motions of the boundary nodes. The resulting simulation is performed using the parameters shown in Table 2. Both geometries were printed in the same orientation as the structure in Section 2.1. by positioning the structure parallel to the base plate of the printer, with each additional layer extruding the geometry into the third direction in space. 

The first sample is the anti-tetrachiral structure attached to circular plates on top and bottom with a diameter of 54.8 mm and thickness of 3 mm. The structure and both plates were printed together as one sample. This choice circumvented the need for additional means of attachment and prevented weak links in the sample. The plates fit in circular fixtures manufactured using an aluminum alloy. The fixtures are mounted onto the compression platens of a Zwick Z100 testing machine and are designed to limit the motion of the sample perpendicular to the loading direction. Since the structure was attached to the platens, beams on top and bottom were constrained in their motion. This constraint resulted in a different deformation field and strength behavior compared to the experiment used for calibrating the model. Simulation results of the compression test with constrained perpendicular motion showed good agreement with the experimental results at small strains. Both force and deformation started to deviate at 17% compressive strain (see Figure 8). The percentage difference between simulation and experiment is listed in Table 3.

The deviation may have different causes. On the one hand, due to the changed CAD geometry, the structure had to be printed in a different orientation, which may have resulted in different imperfections and symmetry issues. This may lead to slightly different deformation behavior, lower strength, and earlier onset of failure. 

It was noted that the agreement between simulation and experiment for this loading condition was slightly worse than for the experiment on which the model was calibrated. However, the results may be considered acceptable for many applications where only compressive strain until the densification regime is sought. The mean relative error between experiments was 5%. Due to high repeatability and similar simulation results, only one of the experimental and simulation results is shown.

The second sample was analyzed in a 3-point-bending test. The corresponding setup is shown in Figure 9. The load is applied by three Delrin wedges, designed to apply load to certain points of the structure. The wedges mounted onto the compression platens of the Zwick Z100 testing machine used purpose-built attachments made out of an aluminum alloy. This experimental setup created a unique deformation behavior as a result of a combined compressive and tensile stress configuration. This configuration allows the simulation to be challenged under more complex loading conditions combined with a varying number of unit cells. 

Results of the 3-point-bending test and the corresponding simulation are shown in Figure 10, with the errors between experiment and simulation reported in Table 4. Note that the quantity “compressive strain” refers to the vertical displacement of the upper wedge, normalized by the initial sample height. Even under significantly more complex loading conditions, the simulation was capable of realistically replicating the corresponding experimental results. The transferability of parameters for one choice of the unit cell across different loading conditions is thus proven. Furthermore, a transferability of parameters was also shown for varying numbers of unit cells within a lattice and/or varying overall dimensions of a structure.

#### 3.2.2. Transferability to a Similar Unit Cell with Different Symmetry Elements

This section investigates whether the calibrated model parameters can be transferred to a different unit cell geometry. This case is different from the investigations of Section 3.2.1. where transferability to different loading conditions with the same unit cell geometry was considered. 

An alternative structure was obtained by rotating the unit cell in Figure 2 by 22.5° relative to the periodic lattice structure. The result was the tetrachiral structure shown in Figure 11. In terms of the slenderness of the beams and their connectivity, this structure was very similar to the anti-tetrachiral structure used for calibrating the model. Therefore, at least a limited predictive capability of the simulation model was expected. However, the absence of mirror-plane symmetry elements implies that the lateral forces (relative to the direction of compression) would not cancel out. As a result, the structure bulges or collapses layerwise. This mechanism makes the numerical prediction of the deformation significantly more challenging, as structural collapse has to be captured. The structure was printed in the same orientation as that in Section 2.1. by positioning the structure parallel to the base plate of the printer, with each additional layer extruding the geometry into the third direction in space. 

The structure was manufactured and analyzed experimentally in the same manner as for the preceding results. It was found that the simulation parameters are not transferable to the different unit cell geometry, as shown in Figure 12. A detailed description of the difference between simulation and experiment is listed in Table 5. The DIC analysis revealed significant geometrical differences between the anti-tetrachiral and tetrachiral structures.

To obtain a working simulation model, a new set of parameters was calibrated, which is reported in Table 6. With these new parameters, the simulation managed to reproduce the deformation field with a displacement error of 5.8%, despite the highly non-homogenous, layerwise collapse of the structure, see Figure 13. The simulated force response deviated from its experimental equivalent by 8.7%, correctly predicting the stiffness and oscillatory behavior in strength.

Similar to the specimen in Section 3.2.1, an alternative sample was designed by attaching two Ø 54.8 mm plates to the structure. This configuration constrained the motion of the beams on top and bottom surfaces perpendicular to the loading direction. The sample was quasi-statically compressed and analyzed using FE-based DIC. The structure was simulated using the parameters of Table 6.

In this case, the simulation again accurately replicated the deformation field up to 16% compressive strain, similar to its anti-tetrachiral equivalent in Section 3.2.1. (see Figure 14). The errors between experiment and simulation are reported in Table 7. It is observed that a transferability of parameters is achieved for one choice of the unit cell across different loading conditions. However, the simulation parameters cannot be transferred between similar yet slightly different unit cell geometries.

## 4. Conclusions

This work introduces a novel phenomenological method to simulate quasi-2D lattice structures, which circumvents many of the problems associated with a direct discretization of such structures. An LPBF-manufactured lattice structure is made of many interconnected small and slender beams. Each beam consists of a solid core coated with a rough surface layer. Only the solid core contributes to the stiffness of the structure, and the rough surface only contributes to the strength at large compression ratios within the densification regime. Beam elements with varying beam thicknesses are utilized to discretize the structure. To account for the rough surface, a contact offset is introduced. FE-based DIC is used to measure displacement fields. The displacement fields, in combination with other experimental data, are used to calibrate the model parameters. The usage of global DIC is critical in the calibration process. In comparison to local DIC, global DIC can utilize regularization, which yields a kinematically correct displacement field.

The final simulation agrees well with the experimental structural deformation and strength. Comparing calibrated parameters to each other confirms the observation about the composition of the printed structure. Additional experiments with alternative loading conditions confirm the ability of the simulation to predict deformation and strength under varying stress states and boundary conditions. It is noteworthy that the beam model accomplishes recreating complex experimental structural deformation, despite a significant simplification of the simulation, defined by only three parameters in total (i.e., two beam thicknesses and a contact offset). 

The method shows good transferability. The reason is the direct coupling of the parameters with the geometry of the unit cell. As long as the unit cell and the manufacturing setup stay unchanged, the unit cell can be used to predict structural behavior under varying loading conditions. Furthermore, a reliable prediction can also be made in the case of varying numbers of unit cells within a lattice and/or varying overall dimensions of a structure. Therefore, a parameterized unit cell can aid in speeding up the design process of structural components, e.g., energy absorbers.

The approach presented herein can be applied to any structure, as it relies only on the minimization between simulation outcome and experimental observables. However, it is computationally only efficient for structures with a high degree of symmetry. It is particularly useful for slender, periodic quasi-2D structures that can be discretized using beam elements. This class of structures includes the majority of additively manufactured lattice structures. It was shown that the approach yields simulations that are faithful for intermediate compressive strains up to approximately 20%. At higher strains, in the region of densification, the predictability is reduced due to failure of the structure and complex contact interactions. Both points require further investigation.

Future work will extend the approach shown here to full 3D structures. In that case, volumetric image data need to be obtained from in-situ X-ray CT experiments in combination with Digital Volume Correlation (DVC) rather than DIC. The method introduced herein can easily be extended into the third dimension, as it only requires the addition of a third direction in space in Equation (2). 

However, this approach yields new challenges. First, a specific setup is required, which allows for measuring force under compression in combination with X-ray CT imaging. Those setups are often limited to a maximum sample size and a maximum force, which restricts the geometry of the structure and choice of the base material. This makes the transferability of the results even more important, as it presents an opportunity to analyze the behavior of a smaller part of a structure, which can be used to predict the behavior of its bigger equivalent. Furthermore, the structure has to be compressed in stages, with each stage being held for the duration of a CT scan. Certain materials, if loaded beyond the point of yield, show an increase in strain without experiencing an increase in stress. Structures printed using those materials are not suitable for this kind of analysis, as the deformation would change over the duration of the CT scan. 

A conclusive set of deformation data requires multiple images. Dependent on the resolution, each CT image can contain up to several GB of data and takes hours to obtain. The total number of images and, therefore, data points per sample is therefore dependent on available time and storage capacity. Furthermore, computational resources required for the DVC analysis are not only dependent on the resolution of the CT scans but also on the resolution of the finite element mesh used as a base. Therefore, the approach requires a delicate balance between computational resources, time management, and information content.

## Figures and Tables

**Figure 1 materials-15-04490-f001:**
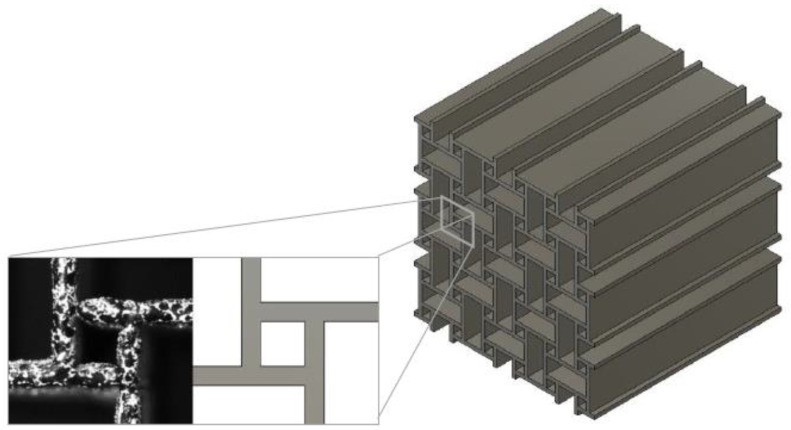
Difference between an LPBF printed anti-tetrachiral unit cell made of Scalmalloy^®^ and the corresponding CAD geometry. In the printed geometry, variations in strut thickness are observed compared to their CAD counterpart.

**Figure 2 materials-15-04490-f002:**
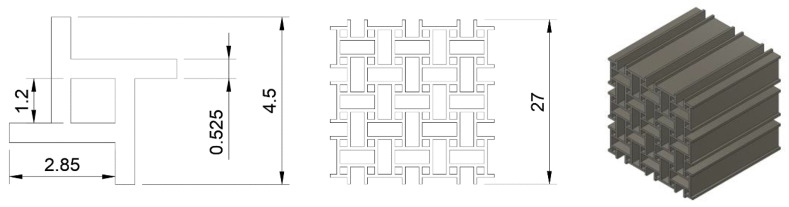
Nominal geometry of the unit cell. The anti-tetrachiral structure is a result of reflective symmetry in two directions in space. The geometry is extruded in the third direction in space, yielding a CAD model suitable for printing. All dimensions are in mm.

**Figure 3 materials-15-04490-f003:**
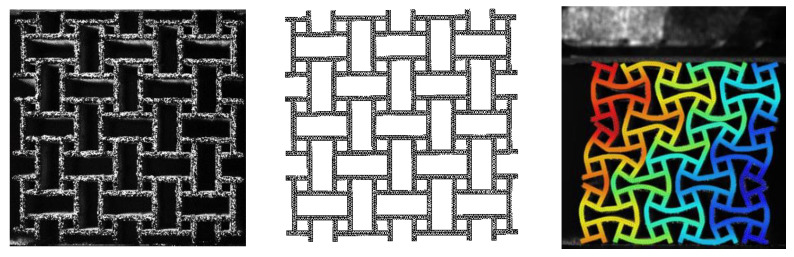
On the left is the initial image of the lattice structure. The center shows the mesoscale FE-mesh based on the nominal CAD geometry. The orientation and position of the mesh are based on the initial state of the structure. On the right, the structure was deformed due to a compressive load. The FE-mesh overlaying the structure allows its mesoscale behaviour to be analyzed.

**Figure 4 materials-15-04490-f004:**
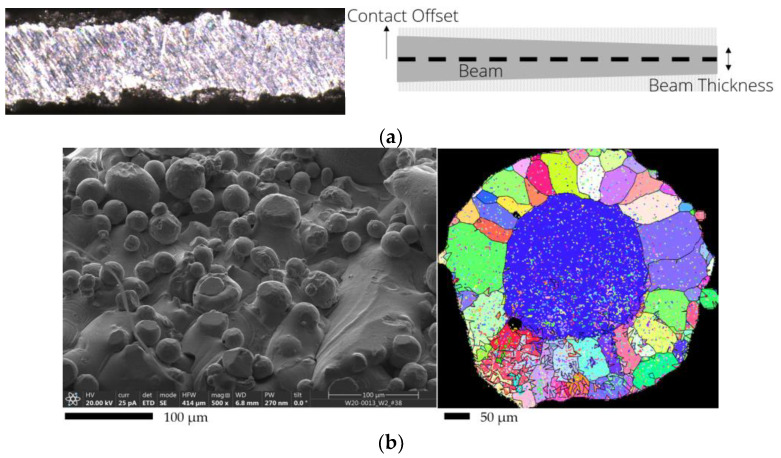
(**a**) Cross-section of an LPBF printed strut, manufactured using Scalmalloy^®^. The corresponding simulation model depicts the solid core surrounded by a rough surface layer. The structural component is part of the structure described in Section 2.1. (**b**) Scanning Electron Microscope (SEM) image of the surface and Electron Backscatter Diffraction (EBSD) grain orientation analysis of an LPBF printed Ø 0.6 mm circular tensile specimen. The specimen was printed perpendicular to the base plate of the printer. The image on the left shows the rough surface layer in more detail, consisting of a combination of unmolten metal particles. The core of the beam shows chunks with different grain orientations.

**Figure 5 materials-15-04490-f005:**
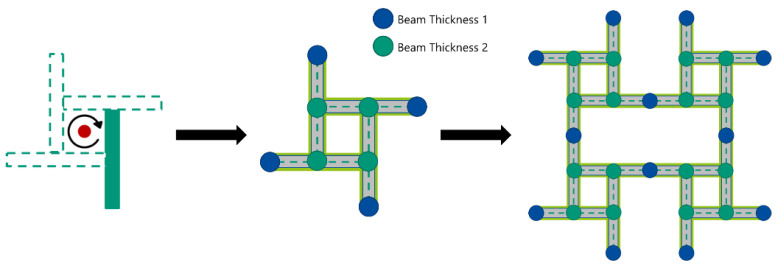
C_4_-rotational symmetry in the unit cell results in groups of nodes with the same geometrical and physical properties. This reduces the number of parameters significantly. Combined with reflective symmetry, the anti-tetrachiral structure is created without increasing the total number of parameters.

**Figure 6 materials-15-04490-f006:**
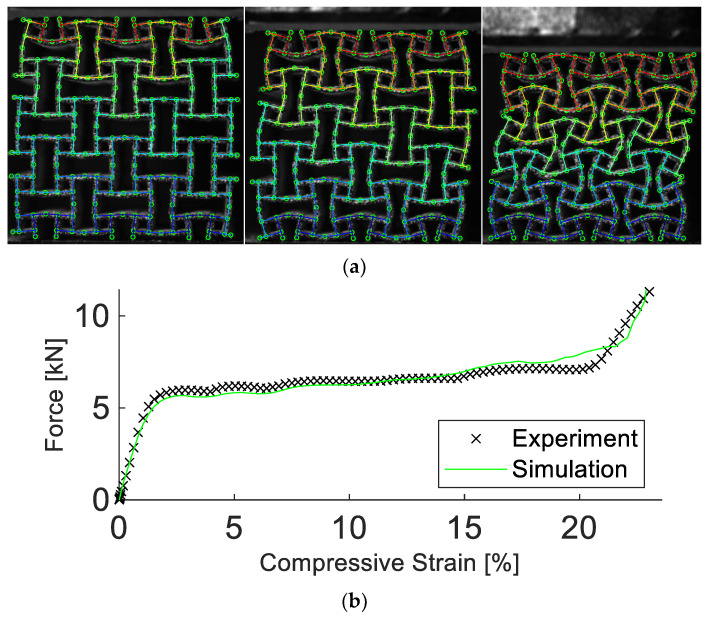
Experimental and simulation results of the anti-tetrachiral structure. Results in (**a**) show the experimental images, visualized as a grey speckle pattern, overlayed by the simulation at 5%, 10%, and 20% compressive strain. In (**b**), the corresponding experimental and simulated force vs. compressive strain are depicted. The simulation manages to replicate the deformation and strength of the structure.

**Figure 7 materials-15-04490-f007:**
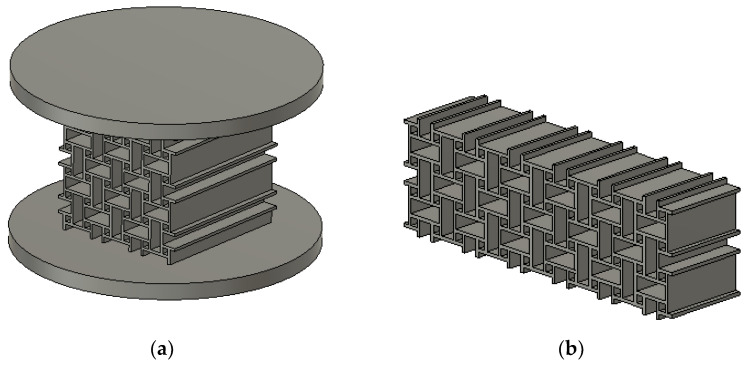
Samples of anti-tetrachiral structure for different types of experiments. Sample in (**a**) was subjected to a compressive load under rigid boundary conditions, while in (**b**), it was subjected to a 3-point-bending test. The resulting loading conditions are different from the situation used for calibration of the model parameters and serve to investigate the predictability of the model for different loading conditions.

**Figure 8 materials-15-04490-f008:**
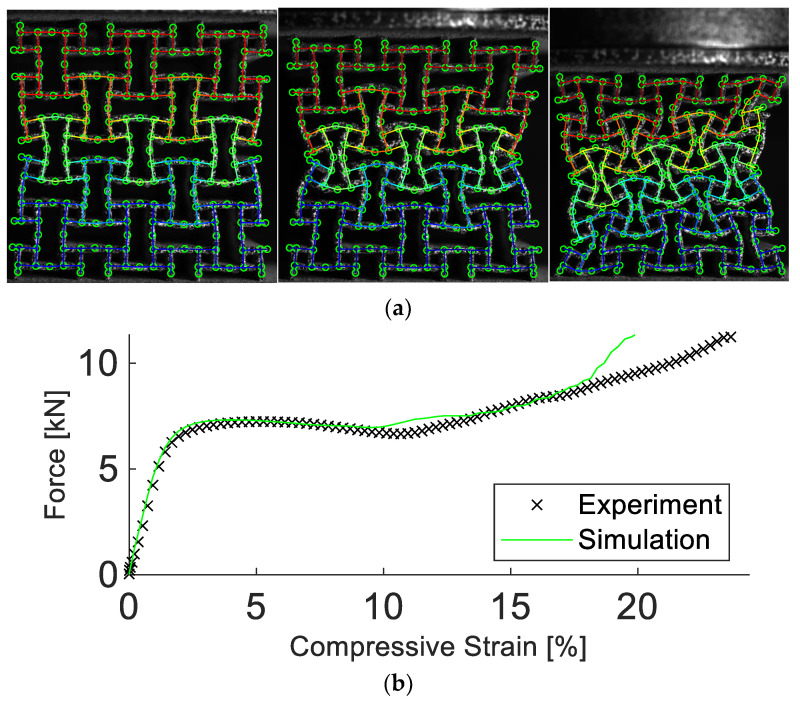
Experimental and simulation results of the anti-tetrachiral structure constrained perpendicular to the loading direction. Results in (**a**) show the deformation at 5%, 10%, and 20% compressive strain. In **(b**), the experimental and simulated force vs. compressive strain are depicted. The simulation accurately predicts the deformation and strength of the structure. The simulation results start to deviate from the experiment when the structure is compressed significantly beyond 17%.

**Figure 9 materials-15-04490-f009:**
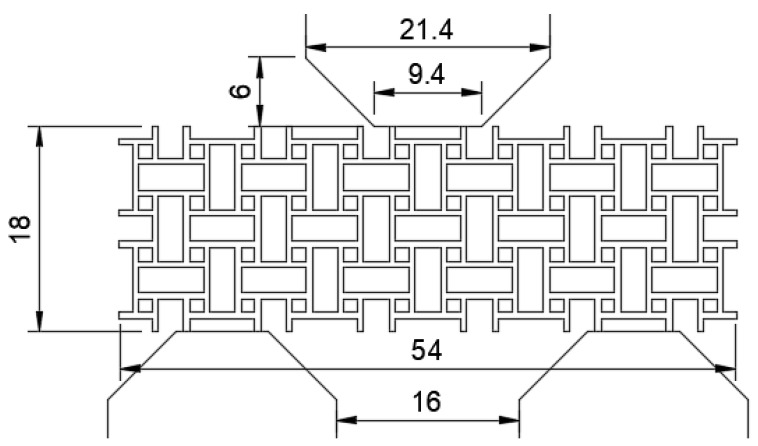
Setup of the 3-point-bending test. The sample was placed between three Delrin wedges. The geometry of the wedges was specifically designed to apply load to certain points of the structure. The wedges were mounted onto the compression platens of a Zwick Z100 testing machine using purpose-built attachments. All dimensions are in mm.

**Figure 10 materials-15-04490-f010:**
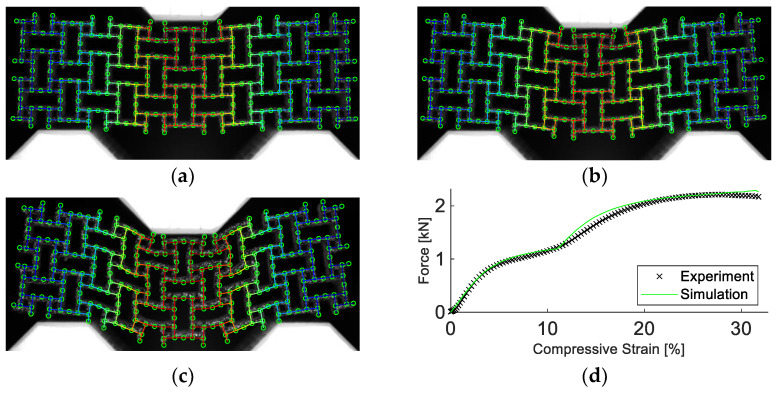
Results of the 3-point-bending test and the corresponding simulation. (**a**) Deformation at 5% compressive strain. (**b**) Deformation at 10% compressive strain. (**c**) Deformation at 20% compressive strain. (**d**) Experimental and simulated force vs. compressive strain. The simulation accurately predicts the deformation and force for the full duration of the experiment.

**Figure 11 materials-15-04490-f011:**
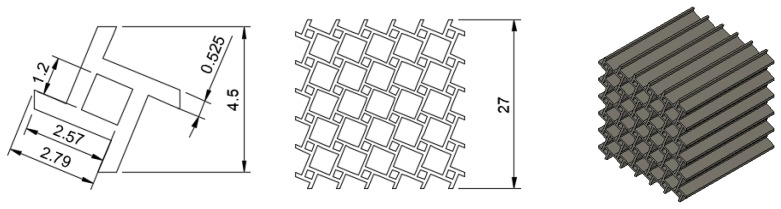
Nominal tetrachiral unit cell geometry. The tetrachiral structure is the a result of repeated duplication in two directions in space. The geometry was extruded in the third direction in space, yielding a CAD model suitable for printing. All dimensions in mm.

**Figure 12 materials-15-04490-f012:**
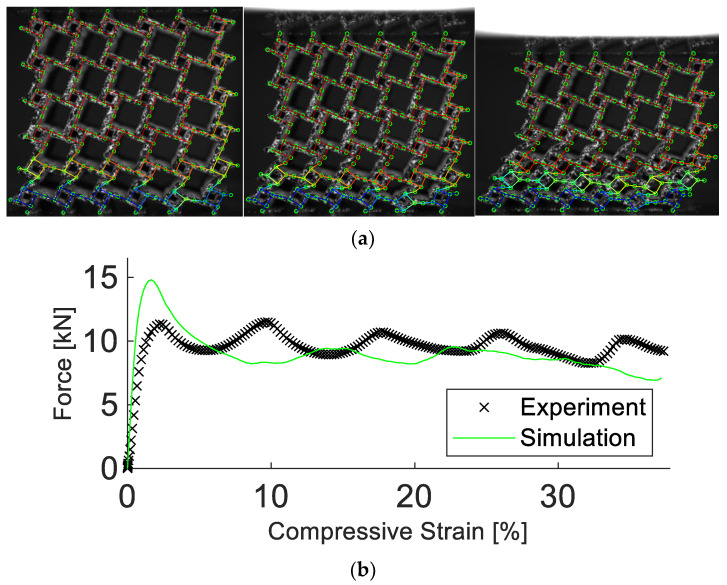
Experimental and simulation results of the tetrachiral structure. Results in (**a**) show the deformation at 5%, 10%, and 20% compressive strain. In (**b**), the experimental and simulated force vs. compressive strain are depicted. The simulation starts to deviate from the experiment at a low compressive strain.

**Figure 13 materials-15-04490-f013:**
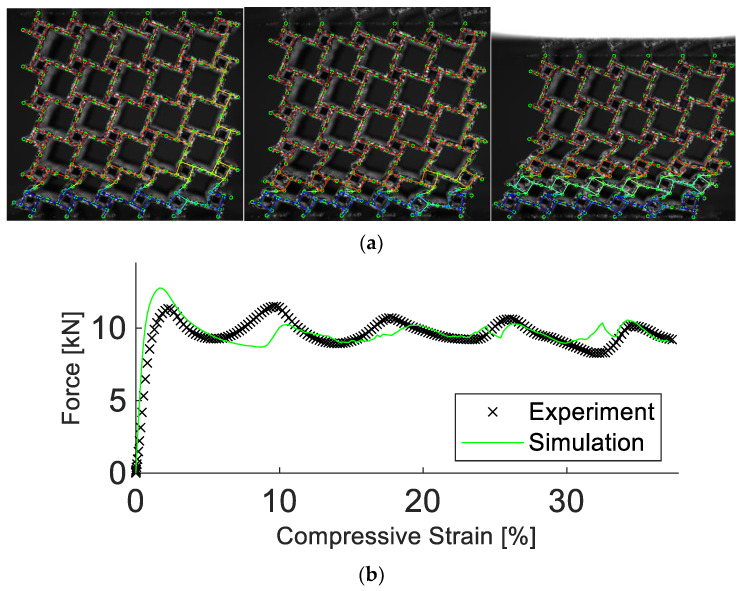
Experimental and simulation results of the tetrachiral structure with a new parameter set. Results in (**a**) show the deformation at 5%, 10%, and 20% compressive strain. In (**b**), the experimental and simulated force vs. compressive strain are depicted. The simulation realistically replicated the results obtained in the experiment.

**Figure 14 materials-15-04490-f014:**
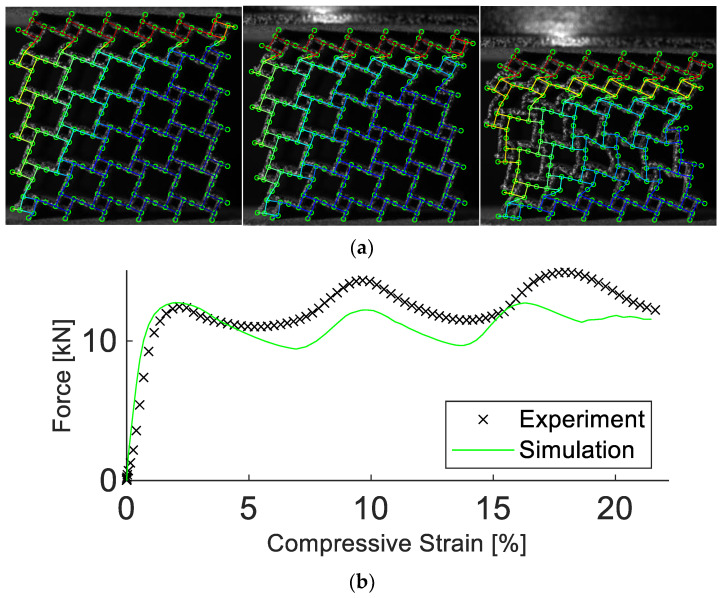
Experimental and simulation results of the tetrachiral structure constrained perpendicular to the loading direction. Results in (**a**) show the deformation at 5%, 10%, and 20% compressive strain. In (**b**), the experimental and simulated force vs. compressive strain are depicted. The simulation predicts stiffness and yield of the structure, deviating in deformation beyond a compressive strain of 16%, similar to its anti-tetrachiral equivalent.

**Table 1 materials-15-04490-t001:** DIC Hardware and Software Parameters.

Software	Correli 3.0
Camera	Basler ace 2
Definition	4096 × 3000 px
Acquisition Framerate	1 fps
Pixel Resolution	11.3 μm
Pattern	Sprayed black and white speckle
Characteristic Pattern Size	15–30 px/0.17–0.34 mm
Grey Level Amplitude	8 bit
Element Type	T3
Mean Element Size	24 px/0.27 mm

**Table 2 materials-15-04490-t002:** Calibration results for the compression test.

Beam Thickness 1	0.53 mm
Beam Thickness 2	0.28 mm
Contact Offset	0.314 mm (0.628 mm total)
Displacement Error	5%
Force Error	3.4%
Total Error	4.2%

**Table 3 materials-15-04490-t003:** NRMSE for simulation of compression test with constrained perpendicular motion up to 23.6% compressive strain.

Displacement Error	Force Error	Total Error
14.6%	9.5%	12.0%

**Table 4 materials-15-04490-t004:** NRMSE for simulation of 3-point-bending test up to 28.2% compressive strain.

Displacement Error	Force Error	Total Error
16.9%	4.5%	10.7%

**Table 5 materials-15-04490-t005:** NRMSE for simulation of compression test of tetrachiral structure using parameters obtained from the anti-tetrachiral structure up to 37.3% compressive strain.

Displacement Error	Force Error	Total Error
24.8%	15.2%	20.0%

**Table 6 materials-15-04490-t006:** Calibration results tetrachiral structure.

Beam Thickness 1	0.46 mm
Beam Thickness 2	0.355 mm
Contact Offset	0.277 mm (0.554 mm total)
Displacement Error	5.8%
Force Error	8.7%
Total Error	7.2%

**Table 7 materials-15-04490-t007:** NRMSE for simulation of compression test with limited perpendicular motion tetrachiral structure up to 21.6% compressive strain.

Displacement Error	Force Error	Total Error
19.7%	12.1%	15.9%

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
