# Peer review of "2D Numerical Simulation of Auxetic Metamaterials Based on Force and Deformation Consistency"

_materials, 2022, doi:10.3390/ma15134490_

Round 1

Reviewer 1 Report

  1. The work presents network structures produced by LPBF, and the aim of the work is a simulation model showing the deformation of the network structure. The model and comparison of strain results for the experiment and FE simulation are shown.

    The detailed comments are as follows:

    1. Please explain all abbreviations and markings at work, e.g. SEM, TF, CPU, and others.
    1. Provide more information about numerical modeling. What program (software) was used, number of finite elements, etc.?
    2. What was the repeatability of the results?
    1. 4 - picture 4b right, scale not very readable - enlarge.
    2. 9 - in the caption, please specify the unit of the given dimensions.

    The strength of the work is the topic taken up and the presented research and simulation results.

    The weakness of the work is the lack of elaboration of the details of the presented issues and some inaccuracies in the work.

Author Response

Thank you for your comments. We gladly implemented the suggestions as follows:

  1. “Please explain all abbreviations and markings at work, e.g. SEM, TF, CPU, and others.”

    We went through the manuscript and added explanations to abbreviations.

  2. “Provide more information about numerical modeling. What program (software) was used, number of finite elements, etc.?”

    Thank you for pointing that out. We added more information about the simulation. The information is noted in lines 237-239.

  3. “What was the repeatability of the results?”

    We corrected this oversight by noting a relative error between our results, which was 5 %.

  4. “4 - picture 4b right, scale not very readable - enlarge.”

    As you suggested, we reviewed both images in Figure 4b and added a more readable scale.

  5. “9 - in the caption, please specify the unit of the given dimensions.”

    Thank you for this remark. We added the missing dimensions to Figure 9.

Reviewer 2 Report

Main contribution:

This paper describes a novel method for simulating metallic lattice structures created by laser powder bed fusion (LPBF). To obtain a simulation model calibrated with experimental data and global Digital Im-age Correlation, a phenomenological approach is investigated (DIC).

Strength:

This paper introduces a novel phenomenological approach to simulation of quasi-2D latticestructures. An LPBF printed structure is made up of many small, interconnected beams. Each beam is made up of a solid core that is coated with a rough surface layer. Only the solid core contributes to the structure's stiffness, and the rough surface contributes to strength only at high compression ratios within the densification regime. The structure is discretized with beam elements of varying thicknesses and a contact offset to account for the rough surface. The model parameters are calibrated using force and the full displacement field obtained from FE-based DIC from experimental data.

Weakness:

The formation of metamaterials is not clearly explained. It must clearly mention each metamaterial and provide valid experimental evidence to support each before and after acceptance value.

Figure 14: The experimental and simulated force deviations are very high, and errors greater than 5% are not universally acceptable for reliable counting.

The value of error in Table 5 is greater, and it is not universally acceptable for reliability between experimental and software.

The formulation of metamaterials is not clearly stated anywhere, and the formulation of experimental wt percent is also not mentioned and validated by some characterization methods that are universally acceptable.

Author Response

Thank you for your valuable comments and the time invested in reviewing our manuscript. While we take these remarks seriously, we disagree with part of the statements.

 “Figure 14: The experimental and simulated force deviations are very high, and errors greater than 5% are not universally acceptable for reliable counting.”

We disagree with this point, as it is very challenging to come up with predictive simulations due to the high number of intrinsic defects caused by the manufacturing process. Our approach, even if it is not perfect, yields better results than other, more conventional simulation types. Considering how simplified our model is, and the total error includes contributions from both displacement errors and force errors, it is our opinion that our results lie well within an acceptable limit. For example, conventional simulations of lattice structures using discretization with volume elements such as tetrahedral as reported in the literature, yield errors worse than ours, see e.g. reference [15] of our manuscript. This is, of course, due to the fact that these models do not consider the intrinsic defects due to the manufacturing process such as pores or dimensional errors due to, e.g., the downskin effect.

The value of error in Table 5 is greater, and it is not universally acceptable for reliability between experimental and software.”

We think this was a misunderstanding. In our manuscript, we stated that the values quoted in Table 5 define an upper limit beyond which we do not deem the simulation results acceptable. The conclusion from this was, that parameters cannot be transferred between two similar structural geometries.

The formulation of metamaterials is not clearly stated anywhere, and the formulation of experimental wt percent is also not mentioned and validated by some characterization methods that are universally acceptable.”

Thank you for pointing this out. We have added a definition of mechanical metamaterials to the beginning of the introductions, near lines 32-39.

We are unsure as to what is meant by “… and the formulation of experimental wt percent is also not mentioned …”. Are you referring to “weight percentage”, i.e., the mass density ratio of the 3D printed structure with respect to the solid material density?
We have added this information to line 131 of the revised manuscript.

Reviewer 3 Report

The authors present a methodology with the aim of enhancing the prediction of the mechanical properties of 2D lattice and cellular metamaterials. While there is a considerable degree of merit, and the work could be of interest to the metamaterials and 3D printing communities, the manuscript fails in giving a complete set of details for many important aspects. Hence, the overall recommendation is to attend these major pieces of information missing before it can be further considered. The amendments should be along the following lines:

  1. In the abstract, What do the authors mean with the word “artefacts” ?
  2. At the end of the abstract authors mention that the possible extension into 3D space is given, but this reviewer could only found a sentence at the very end of the conclusions related to this. Is this what the authors call “discussion”?
  3. The literature review is missing. This in order to highlight the contribution of the authors. One is expecting a review of what has been done to predict the mechanical properties of metallic additively manufactured periodic porous structures. Perhaps the inclusion of some reviews on the use of additive manufacturing for the fabrication of cellular and lattice materials will also aid to give a insight into what has been done.
  4. Paragraph from lines 57 to 60 needs references to support what is being said.
  5. How were the samples printed? This is important as anisotropy at the strut level could affect the mechanics if other orientation is used for manufacturing.
  6. Which where the boundary conditions used for their finite element simulations described in Section 2.3?
  7. More information is needed on how the samples in Figure 4 were obtained. Are these part or sections of the metamaterial studied? How were these printed? Por instance figure 4a contains a section of one of the struts of the lattice, was this lying parallel to the horizontal or vertical axis (taking the plane in Fig 3 as reference).
  8. In many sections authors present the figures before these are even mentioned in the text, e.g. Fig 6.
  9. How’s the force obtained from a DIC approach?
  10. Paragraph in Lines 286 to 292, “comparing the resulting parameters” which parameters?
  11. Section 3.2.1 how are the plates for compression mounted to the sample?
  12. Section 3.2.1 if parameters studied are at the unit cell level, and the unit cell is kept unchanged for the other loading conditions, is not this a reason why the transferability results with good agreement? Can the authors extend their discussion along this line?
  13. Instead of “3-point-bend”, the term “3-point bending” is more used in the literature.
  14. Dimension in Fig. 9?

Author Response

Thank you for your very detailed comments and suggestions. We went through the manuscript and addressed each point.

  1. “In the abstract, What do the authors mean with the word “artefacts” ?”

    We are sorry about the confusion. In this context, print artefacts refer to imperfections or defects within the printed object caused by either uncertainties (scatter) or systematic errors within the printing process. We have changed the abstract at line 17 to be more specific.

  2. “At the end of the abstract authors mention that the possible extension into 3D space is given, but this reviewer could only found a sentence at the very end of the conclusions related to this. Is this what the authors call “discussion”?”

    Thank you for pointing this out. We expanded on this in the conclusion by adding multiple paragraphs, adding how we want to approach the extension in the third dimension and what challenges we will have to face.

  3. “The literature review is missing. This in order to highlight the contribution of the authors. One is expecting a review of what has been done to predict the mechanical properties of metallic additively manufactured periodic porous structures. Perhaps the inclusion of some reviews on the use of additive manufacturing for the fabrication of cellular and lattice materials will also aid to give a insight into what has been done.”
  4. “Paragraph from lines 57 to 60 needs references to support what is being said.”

    Thank you for suggesting this improvement, We expanded the introduction by using more references for our statements. Furthermore, we added a more detailed review on what already has been done in the department of metamaterial simulations, providing a more detailed overview of the literature that is significant for our work.

  5. “How were the samples printed? This is important as anisotropy at the strut level could affect the mechanics if other orientation is used for manufacturing.”

    You are right, the orientation is very important. We corrected this oversight by adding to each structure how it was oriented in the printing process.

  6. “Which where the boundary conditions used for their finite element simulations described in Section 2.3?”

    We addressed the boundary conditions in lines 230-236.

  7. “More information is needed on how the samples in Figure 4 were obtained. Are these part or sections of the metamaterial studied? How were these printed? Por instance figure 4a contains a section of one of the struts of the lattice, was this lying parallel to the horizontal or vertical axis (taking the plane in Fig 3 as reference).”

    Thank you for pointing this out. We supplemented Figure 4 with more details (orientation, structural component,…). However, we have to add that this figure is meant to show typical defects, that can be expected from this type of manufacturing, independent of the orientation or material used.

  8. “In many sections authors present the figures before these are even mentioned in the text, e.g. Fig 6.”

    We do this due to personal preference. This way the reader can check the image first, before it is described in detail in the text. We like to add, that, in our experience, the final figure positioning in the published layout is often controlled by other factors outside of our reach.

  9. “How’s the force obtained from a DIC approach?”

    We think there is a misunderstanding. The force was measured via a load cell when compressing the specimens using a Zwick Z100 testing machine. The DIC analysis is computed separately. Force measurement and image acquisition are triggered by two signals at the same time, so the DIC result can be seamlessly interconnected with the measured force.

    We see that our initial statement (“… in addition to force, DIC was used …”) was misleading, and we rephrased this statement at line 138.

  1. “Paragraph in Lines 286 to 292, “comparing the resulting parameters” which parameters?”

    Thank you for pointing out, that this part might be a bit confusing. We changed the sentence to “all resulting parameter values (beam thickness 1 and 2 and contact offset)” to be more precise.

  2. “Section 3.2.1 how are the plates for compression mounted to the sample?”

    We corrected this oversight by adding “The structure and both plates were printed together as one sample. This circumvents the need of additional means of attachment and prevents weak links in the sample.” to the paragraph.
  1. „Section 3.2.1 if parameters studied are at the unit cell level, and the unit cell is kept unchanged for the other loading conditions, is not this a reason why the transferability results with good agreement? Can the authors extend their discussion along this line?“

    This is a good point. Yes, the parameters calculated are directly coupled with the unit cell. As you stated, this is the reason for the good transferability. With our results, we want to prove that knowing the parameters of the unit cell is enough to setup simulations with different numbers of unit cells and different boundary conditions and obtain realistic result. The unit cell could be used in different contexts and aid the design of components like e.g. energy absorbers. This was not clearly stated in the original manuscript, so we added this in the conclusion.
  1. “Instead of “3-point-bend”, the term “3-point bending” is more used in the literature.”

    Similar to point 6., the term “3-point-bend” was used as a personal preference. We changed it to “3-point bending”.
  1. “Dimension in Fig. 9?”

    Thank you for this observation. We added the missing dimensions to Figure 9.

Round 2

Reviewer 3 Report

The answer to comment 8. “In many sections authors present the figures before these are even mentioned in the text, e.g. Fig 6.” is not satisfactorily provided.
Authors mentioned it is their personal preference, figure should never appear before they are mentioned, at least in the review process. They argued that they like the readers to see the figure first, yes but without even knowing what we are looking at? is authors want to send the atention to the figure, they most do so in the writing. additionally, They mentioned that the final position of the figure is controlled by the journal, which is true, but this version is not the final one, so they do have control on where to put it. 

Please check the manuscript for errors in the compiling process (line 187)

Author Response

Point 1:

"The answer to comment 8. “In many sections authors present the figures before these are even mentioned in the text, e.g. Fig 6.” is not satisfactorily provided.
Authors mentioned it is their personal preference, figure should never appear before they are mentioned, at least in the review process. They argued that they like the readers to see the figure first, yes but without even knowing what we are looking at? is authors want to send the atention to the figure, they most do so in the writing. additionally, They mentioned that the final position of the figure is controlled by the journal, which is true, but this version is not the final one, so they do have control on where to put it. "

Thank you for your remark. We revisioned the manuscript and adjusted the position of each figure accordingly.

Point 2:

"Please check the manuscript for errors in the compiling process (line 187)"

Thank you for pointing this out. We will check for errors before submitting.